# Discovery of Novel 1,2,3-triazole Derivatives as IDO1 Inhibitors

**DOI:** 10.3390/ph15111316

**Published:** 2022-10-25

**Authors:** Xixi Hou, Xiaoqing Gong, Longfei Mao, Jie Zhao, Jianxue Yang

**Affiliations:** 1The First Affiliated Hospital and College of Clinical Medicine of Henan University of Science and Technology, Luoyang 471003, China; 2College of Chemistry and Chemical Engineering, Lanzhou University, Lanzhou 730000, China; 3School of Chemistry and Chemical Engineering, Henan Normal University, Xinxiang 453007, China; 4School of Nursing, Henan University of Science and Technology, 263 Kaiyuan Road, Luoyang 471003, China

**Keywords:** indoleamine 2,3-dioxygenase 1, 1,2,3-triazole, molecular docking, quantum mechanical studies

## Abstract

Indoleamine 2,3-dioxygenase 1 (IDO1) has received much attention as an immunomodulatory enzyme in the field of cancer immunotherapy. While several IDO1 inhibitors have entered clinical trials, there are currently no IDO1 inhibitor drugs on the market. To explore potential IDO1 inhibitors, we designed a series of compounds with urea and 1,2,3-triazole structures. Organic synthesis and IDO1 enzymatic activity experiments verified the molecular-level activities of the designed compounds, and the IC_50_ value of compound **3a** was 0.75 μM. Molecular docking and quantum mechanical studies further explained the binding mode and reaction potential of compound **3a** with IDO1. Our research has resulted in a series of novel IDO1 inhibitors, which is beneficial to the development of drugs targeting IDO1 in numerous cancer diseases.

## 1. Introduction

Immuno-oncology is an important form of tumor treatment, along with surgery, radiotherapy, and chemotherapy [1]. It works by activating and mobilizing the body’s own immune system through positive regulation of the immune system, subsequently achieving the capture and clearance of tumor cells [2]. At present, immuno-oncology is largely performed in clinical situations via immune checkpoint inhibition, tumor vaccine therapy, and adoptive cell therapy. Immune checkpoint inhibitors (PD1/PD-L1) have become first-line treatments for advanced non–small-cell lung cancer and melanoma [3]. As another important immune checkpoint, IDO1 overexpression, is observable in many types of tumor cells, such as those of gastric cancer, breast cancer, and brain cancer. IDO1 is a monomeric protein enzyme containing heme [4]. Its main biological function is to catalyze the oxidative cleavage of the L-tryptophan (L-Trp) indole ring to produce kynurenine (KYN). The consumption of L-Trp blocks the proliferation of T-cells and promotes the differentiation of T-cells into regulatory T-cells [5]. KYN metabolites can combine with aryl hydrocarbon receptors (AhRs) to activate signaling pathways, enhancing immune tolerance and leading to immune escape. The toxicity of accumulated metabolites also causes T-cell apoptosis. Consequently, IDO1 inhibitors have attracted the attention of scientists as potential cancer therapeutic drugs. Currently, a few drugs have entered the clinical trial stage, such as 4-PI (Figure 1,**1**), NLG919 (Figure 1,**2**), Epacadostat (Figure 1,**3**), and Amg-1 (Figure 1,**4**) [6]. Amg-1 is a selective IDO1 inhibitor with better performance than TDO2 and IDO2. The IC_50_ of Amg-1 was found to be 3.0 μM by the Bridge-It tryptophan fluorescence assay. Our group further analyzed the crystal structure of an Amg-1/IDO1 complex (PDB: 4PK5) and uncovered additional characteristics affecting enzyme recognition by small molecules [7]. Pockets A and B are formed by ligand-induced conformational rearrangement of the residue binding sites located in the catalytic cleft (Figure 2). Of these, pocket A is largely composed of heme and the amino acid residues Tyr126, Cys129, Val130, Phe163, Phe164, Gly262, and Ala264, while pocket B is located at the entrance of the catalytic site and is composed of the residues Phe226, Phe227, Arg231, Ile354, and Leu384. IDO1 inhibitors specifically occupy either pocket A alone or both pockets [8]. This observation confirms the presence of conformational changes in the catalytic site induced by multiple ligands and supports the molecular recognition of enzymes toward inhibitors with different structures. Aryl, halogen-substituted aryl, or heteroaromatic groups (such as indole) are most often chosen to occupy pocket A, while pocket B contains a mixture of hydrophilic (Arg231) and hydrophobic (Phe226) residues [9]. Hence, it is important in the development of small-molecule inhibitors to understand the structural composition of both catalytic pockets of the IDO1 enzymes and their interaction with IDO1 inhibitors.

Urea compounds are used as enzyme inhibitors and lead compounds to develop biomimetic peptides because of their peptide bonds (CONH group) and biological activities akin to those of peptides [10]. These characteristics render them important chemicals in pesticides and medicine, such as herbicides [11], insecticides, bactericides [12,13], and plant-growth regulation [14]. The ability of peptide bonds in urea compounds to combine with enzymes through hydrogen bonding could be manipulated to synthesize cyclin-dependent kinases to block the cell cycle [15]. At present, urea compounds are largely obtained by the direct reaction of isocyanate compounds and amine compounds [16]. Our group performed condensation reactions on differentially substituted phenyl isocyanates and alkyne-terminated amines and obtained, through click chemistry, compounds with urea and 1,2,3-triazole structures. 

Through comprehensive literature research, our group designed and synthesized a series of novel compounds modeled after the binding mechanism between Amg-1 and the IDO1 enzyme, replacing the oxadiazole structure with the 1,2,3-triazole structure and the hydroxylamine structure with a urea structure. We selected three types of functional groups: *m*-phenyl group, *p*-phenyl group, and methyl group as Linkers to connect urea and 1,2,3-triazole blocks respectively. The reaction routes of the three series of compounds (**3a**–**3i**, **5a**–**5f**, **7a**–**7f**) are shown in Figure 3 and Table 1. The action conditions of the step were convenient and easy to control. Different substituents (R^1^, R^2^, and R^3^) were present on the benzene ring of phenyl isocyanate and benzyl structures. The effect of linker substituents and R^1^, R^2^, and R^3^ on IDO1 inhibition were compared.

## 2. Results and Discussion

### 2.1. IDO1 Inhibition Study

Through literature surveys, we employed the Hela cell-based functional assay to study the IDO1 inhibition activities of the designed compounds. The IC_50_ value of BMS-986205 as a positive control was 0.62 nM (data not shown), which is consistent with the results previously reported in the literature (IC_50_ = 0.5 nM) [2]. As shown in Table 2, the activities of compounds **3a**–**3i** against IDO1 were all below 10 μM, of which **3a** reached 0.75 μM. The activity of most of compounds **5a**–**5f** and **7a**–**7f** were higher than 10 μM, indicating that the activity of *m*-phenyl as the linking substituent of urea and triazole was better than that of *p*-phenyl and methyl. We further compared the effects of different linking groups on the activity by selecting compounds with the same substituents on the left and right benzene rings. Then three groups of compounds (i.e., **3a**, **5e** and **7f**; **3e**, **5d** and **7e**; and **3g**, **5c** and **7d**) were compared to verify that *m*-phenyl is the better linker substituent in terms of promoting inhibitor activity. It was also found by comparing **3f**, **5b** and **7b** that the three compounds with the same benzene ring substituents on their phenyl isocyanate and benzyl structures but different linker substituents have similar inhibitory activities. This indicates that benzene ring substituents R^1^ = OCH_3_, R^2^ = H, and R^3^ = Br, all give synthesized compounds good inhibitory activities.

### 2.2. Molecular Docking Studies of Compounds ***3a, 5e*** and ***7f***

In order to intuitively analyze the binding mode of active compounds to IDO1, we selected compounds **3a**, **5e** and **7f** with different orders of magnitude activity as model compounds. As shown in Figure 4, the docking scores of compounds **3a**, **5e** and **7f** with IDO1 were −7.812, −6.216 and −7.815, respectively. The 1,2,3-triazolyl group in the structures of compounds **3a** and **7f** was located above the heme and the distance from the ferrous ion in the middle of the heme is 3.33 Å and 3.43 Å, respectively. The binding modes of the compounds are similar to several reported IDO1 inhibitors. The 1,2,3-triazole group in the structure of compound **5e** has a distance of 4.82 Å from the heme ferrous ion, which is far away from the heme iron and has a weaker affinity for IDO1. The docking results indicated that compounds **3a** and **7f** were more capable of coordinating with ferrous ions than **5e**. The 1,2,3-triazole in the structure of compound **7f** is connected to the urea structure through a flexible methylene group, and the steric twist of the methylene group affects the position of compound **7f** in the pocket; thus, the inhibitory activity of compound **7f** on IDO1 is weaker than that of compound **3a**.

### 2.3. HOMO-LUMO and MEP Studies

The HOMO and LUMO orbitals of the molecule and the energetic gap values of the orbital characterized the chemical reactions and kinetic stability. The smaller the energetic gap value is, the less stable the compound is, and it is more likely to interact with the protein [17,18]. The energies of Frontier molecular orbitals of compounds **3a**, **5e** and **7f** are shown in Figure 5 and Table 3. The energy gap values of the three compounds of HOMO-LUMO were, respectively, 4.5762, 4.6692, and 4.9914 eV, indicating that the chemical reactions of the compound were **3a** > **5e** > **7f**. The result showed that compound **3a** is more likely to interact with the IDO1 protein, followed by **5e**; **7f** is the worst. Moreover, the calculated results were in good agreement with the experimental results. Furthermore, the urea group in compounds **3a** and **5e** was located in the HOMO orbitals, which provided electrons for the amino acid interaction at the IDO1 binding pocket.

Molecular electrostatic potential (MEP) can help us obtain important information about the charge density, which is easy to ensure the sites of electrophilic nucleophilic reaction of muons, and also reflect the molecular interactions and physicochemical property characteristics [19,20]. We performed an MEP analysis to gain a better understanding of the electrophilic and nucleophilic reaction sites of compound **3a**. The electrostatic potential surface map of compound **3a** is shown in Figure 6. Different colors of the electrostatic potential surface represent the changing trend of the potential energy, and the increasing trend of the potential energy is red > yellow > green > blue. The negative regions (red and yellow) of the electrostatic potential map of the compound are correlated with electrophilic reactivity, and positive regions (blue) are related to nucleophilic reactivity. The highly negative region of compound **3a** is located at O16 of the urea group, and the highly positive region is located at the NH group involved in the N9 of the urea group. The atoms in these regions may form a hydrogen bond interaction protein with IDO1, which is consistent with the docking results. In addition, the N20 and N21 of 1,2,3-triazole in the structure of compound **3a** are located in the orange region with higher electron density, which can serve as the basic site for heme iron binding [21].

## 3. Materials and Methods

### 3.1. Materials and Chemistry

The 1,2,3-triazole derivatives were synthesized in-house by our research group. All reagents and solvents were obtained from a commercial source (Sinopharm, Beijing, China) and used without further treatment. ^1^H NMR and ^13^C NMR spectra were recorded in DMSO-d_6_ solution with a Bruker 600 spectrometer (Bruker, Karlsruhe, Germany). Chemical shifts (d) were recorded in parts per million. Tetramethylsilane (Acros, Organics, Brussels, Belgium) was used as an internal reference, and coupling constants were expressed in hertz. High-resolution mass spectra (HRMS) measurements were carried out using a Bruker Micro ToF II mass spectrometer (Bruker, Karlsruhe, Germany).

Hela cell line and DMEM mediumand fetal bovine serum were purchased from ATCC (Manassas, VA, USA) commercially, and Recombinant Human IFN-γ was purchased from R&D systems (Emeryville, CA, USA) commercially. The 3.05 Ntrichloroacetic acid, 4-(dimethylamino) benzaldehyde, and acetic acid were purchased from Sigma Aldrich (St. Louis, MO, USA).

### 3.2. General Procedure for Preparation of Compound ***2a*** (The Method Is Suitable for ***2b, 4a, 4b, 6a*** and ***6b***)

To a solution of 4-bromophenyl isocyanate (2.0 g, 0.01 mol) in CH_2_Cl_2_ (100 mL), 3-aminophenylacetylene (1.2 g, 0.01 mol) was addedin one portion. After stirring at room temperature for 2.5 h, the mixture was concentrated, and the residue was purified by column chromatography on silica gel (eluent:PE:EA = 2:1) to make compound **2a** a pale yellow solid.

### 3.3. General Procedure for Preparation of Compound ***3a***

The reaction reagents were compound **2a** (0.32 g, 1.0 mmol) and1-(azidomethyl)-3-methoxybenzene (0.2 g, 1.2 mmol), which were added to 15 mL of mixed solvent (water:tert-butanol = 2:1). The reaction was performed in copper sulfate pentahydrate (0.1 mmol) and sodium ascorbate (0.2 mmol) at 80 °C. The reaction status was monitored by TLC, and after completion of the reaction, the mixture was extracted through dichloromethane (15 mL × 3). The organic phase was combined and washed successively with water and brine, then dried over sodium sulfate and concentrated in vacuo. The desired compound **3a** was isolated by column chromatography (CH_2_Cl_2_/MeOH = 20:1).

***1-(4-bromophenyl)-3-(3-(1-(3-methoxybenzyl)-1H-1,2,3-triazol-4-yl)phenyl)urea* (3a)**. White solid. HR MS (ESI) m/z: calcd for C_23_H_21_BrN_5_O_2_ [M+H]^+^478.0879, found 478.0876. ^1^H NMR (600 MHz, DMSO-*d_6_*) δ 8.82 (s, 2H), 8.60 (s, 1H), 8.03 (s, 1H), 7.45–7.42 (m, 5H), 7.36–7.28 (m, 3H), 6.96 (t, *J* = 2.4 Hz, 1H), 6.92 (d, *J* = 4.2 Hz, 1H), 6.90 (d, *J* = 3.6 Hz, 1H), 5.60 (s, 2H), 3.75 (s, 3H). ^13^C NMR (150 MHz, DMSO-*d_6_*) δ 159.47, 152.39, 146.61, 140.05, 139.10, 137.45, 131.51, 129.97, 129.38, 121.60, 120.19, 120.01, 119.05, 117.85, 114.92, 113.77, 133.50, 113.25, 55.12, 55.92.

***1-(3-(1-(3-methoxybenzyl)-1H-1,2,3-triazol-4-yl)phenyl)-3-(4-methoxyphenyl)urea* (3b)**. White solid. HR MS (ESI) m/z: calcd for C_24_H_24_N_5_O_3_ [M+H]^+^430.1879, found 430.1871. ^1^H NMR (600 MHz, DMSO-*d_6_*) δ 8.70 (s, 1H), 8.60 (s, 1H), 8.48 (s, 1H), 8.03 (t, *J* = 1.8 Hz, 1H), 7.41 (dt, *J*_1_ = 7.8 Hz, *J*_2_ = 1.2 Hz, 1H), 7.39–7.37 (m, 3H), 7.34-7.29 (m, 2H), 6.97 (t, *J* = 1.8 Hz, 1H), 6.92 (dd, *J*_1_ = 8.4 Hz, *J*_2_ = 2.4 Hz, 2H), 6.89-6.87 (m, 2H), 5.61 (s, 2H), 3.75 (s, 3H), 3.72 (s, 3H). ^13^C NMR (150 MHz, DMSO-*d_6_*) δ 159.95,155.00, 153.20, 147.17, 140.93, 137.94, 133.13, 131.62, 130.44, 129.81, 122.04, 120.58, 120.49, 119.18, 118.11, 155.18, 144.47, 114.24, 113.97, 55.65, 55.59, 53.41.

***1-(4-fluorophenyl)-3-(3-(1-(3-methoxybenzyl)-1H-1,2,3-triazol-4-yl)phenyl)urea* (3c)**. Gray solid. HR MS (ESI) m/z: calcd for C_23_H_21_FN_5_O_2_ [M+H]^+^418.1679, found 418.1684. ^1^H NMR (600 MHz, DMSO-*d_6_*) δ 8.77 (s, 1H), 8.70 (s, 1H), 8.60 (s, 1H), 8.02 (t, *J* = 1.8 Hz, 1H), 7.50–7.46 (m, 2H), 7.42 (dt, *J*_1_ = 7.2 Hz, *J*_2_ = 1.2 Hz, 1H), 7.37 (dt, *J*_1_ = 9.0 Hz, *J*_2_ = 1.2 Hz, 1H), 7.34 (d, *J* = 7.8 Hz, 1H), 7.30 (t, *J* = 7.8 Hz, 1H), 7.12 (m, 2H), 6.96 (t, *J* = 1.8 Hz, 1H), 6.91 (dd, *J*_1_ = 8.4 Hz, *J*_2_ = 3.0 Hz, 2H), 5.60 (s, 2H), 3.75 (s, 3H). ^13^C NMR (150 MHz, DMSO-*d_6_*) δ 159.47, 156.58, 152.63, 146.64, 140.22, 137.46, 131.16, 129.96, 129.36, 121.58, 120.07, 120.01, 118.91, 117.78, 115.34, 115.19, 114.85, 113.76, 113.49, 55.12, 52.93.

***1-(3-(1-(3-methoxybenzyl)-1H-1,2,3-triazol-4-yl)phenyl)-3-phenylthiourea* (3d)**. White solid. HR MS (ESI) m/z: calcd for C_23_H_22_N_5_OS[M+H]^+^416.1545, found 416.1546. ^1^H NMR (400 MHz, CDCl_3_) δ 7.91 (s, 1H), 7.82 (s, 1H), 7.79 (s, 1H), 7.68 (s, 1H), 7.45 (d, *J* = 7.2 Hz, 1H), 7.47–7.41 (m, 4H), 7.38 (t, *J* = 7.2 Hz, 2H), 7.30 (t, *J* = 8.0 Hz, 2H), 6.91–6.88 (m, 2H), 6.82 (t, *J* = 1.6 Hz, 1H), 5.53 (s, 2H), 3.79 (s, 3H). ^13^C NMR (150 MHz, DMSO-*d_6_*) δ 179.12, 159.31, 146.35, 137.09, 135.89, 135.00, 131.01, 129.44, 128.95, 128.89, 126.51, 124.57, 123.89, 123.13, 121.23, 119.43, 119.11, 113.46, 112.88, 54.47, 53.42.

***1-(3-(1-(3-bromobenzyl)-1H-1,2,3-triazol-4-yl)phenyl)-3-(4-bromophenyl)urea* (3e)**. White solid. HR MS (ESI) m/z: calcd for C_22_H_18_Br_2_N_5_O [M+H]^+^525.9878, found 525.9868. ^1^H NMR (600 MHz, DMSO-*d_6_*) δ 8.84 (s, 2H), 8.66 (s, 1H), 8.04 (t, *J* = 1.8 Hz, 1H), 7.61 (s, 1H), 7.57–7.55 (m, 1H), 7.46 (s, 3H), 7.44 (dt, *J*_1_ = 7.2 Hz, *J*_2_ = 1.8 Hz, 1H), 7.37–7.36 (m, 3H), 6.87 (s, 1H), 6.65 (s, 1H), 5.66 (s, 2H). ^13^C NMR (150 MHz, DMSO-*d_6_*) δ 152.87, 151.94, 147.16, 139.66, 139.12, 131.99, 131.55, 131.51, 131.23, 128.51, 127.56, 125.39, 122.35, 122.24, 120.65, 118.37, 115.38, 113.73, 52.63, 34.86, 30.89, 21.52.

***1-(3-(1-(3-bromobenzyl)-1H-1,2,3-triazol-4-yl)phenyl)-3-(4-methoxyphenyl)urea* (3f)**. Brown solid. HR MS (ESI) m/z: calcd for C_23_H_21_BrN_5_O_2_ [M+H]^+^478.0879, found 478.0875. ^1^H NMR (600 MHz, DMSO-*d_6_*) δ 8.71 (s, 1H), 8.64 (s, 1H), 8.49 (s, 1H), 8.03 (s, 1H), 7.61 (s, 1H), 7.56–7.55 (m, 1H), 7.38 (s, 1H), 7.36 (d, *J* = 4.2 Hz, 4H), 7.33 (d, *J* = 7.8 Hz, 1H), 6.87 (d, *J* = 9.0 Hz, 2H), 3.72 (s, 3H). ^13^C NMR (150 MHz, DMSO-*d_6_*) δ 154.97, 153.19, 147.24, 140.95, 139.13, 131.55, 131.51, 131.23, 129.83, 127.56, 122.35, 122.21, 120.55, 119.16, 118.14, 115.15, 114.45, 55.64, 52.63, 30.89.

***1-(3-(1-(2-bromobenzyl)-1H-1,2,3-triazol-4-yl)phenyl)-3-(4-bromophenyl)urea* (3g)**. Gray solid. HR MS (ESI) m/z: calcd for C_22_H_18_Br_2_N_5_O [M+H]^+^525.9878, found 525.9885. ^1^H NMR (600 MHz, DMSO-*d_6_*) δ 8.84 (d, *J* = 6.0 Hz, 2H), 8.58 (s, 1H), 8.03 (t, *J* = 1.8 Hz, 1H), 7.71 (d, *J* = 7.8 Hz, 1H), 7.46 (s, 4H), 7.44 (d, *J* = 1.8 Hz, 1H), 7.44-7.42 (m, 1H), 7.40 (dd, *J*_1_ = 6.6 Hz, *J*_2_ = 1.8 Hz, 1H), 7.35 (d, *J* = 8.4 Hz, 1H), 7.33 (dd, *J*_1_ = 7.8 Hz, *J*_2_ = 1.8 Hz, 1H), 7.22 (dd, *J*_1_ = 7.8 Hz, *J*_2_ = 1.2 Hz, 1H), 5.74 (s, 2H). ^13^C NMR (150 MHz, DMSO-*d_6_*) δ 152.88, 146.92, 140.54, 138.57, 135.34, 133.39, 131.99, 131.57, 130.90, 129.88, 128.82, 123.31, 122.52, 120.71, 120.66, 119.59, 118.38, 115.41, 113.73, 53.56.

***1-(3-(1-(2-bromobenzyl)-1H-1,2,3-triazol-4-yl)phenyl)-3-(4-fluorophenyl)urea* (3h)**. Gray solid. HR MS (ESI) m/z: calcd for C_22_H_18_BrFN_5_O [M+H]^+^466.0679, found 466.0673. ^1^H NMR (600 MHz, DMSO-*d_6_*) δ 8.95 (s, 1H), 8.90 (s, 1H), 8.59 (s, 1H), 8.03 (t, *J* = 1.8 Hz, 1H), 7.71 (dd, *J*_1_ = 7.8 Hz, *J*_2_ = 1.2 Hz, 1H), 7.48 (dd, *J*_1_ = 9.0 Hz, *J*_2_ = 4.8 Hz, 2H), 7.43 (t, *J* = 6.6 Hz, 2H), 7.40 (d, *J* = 7.8 Hz, 1H), 7.35-7.33 (m, 2H), 7.22 (dd, *J*_1_ = 7.8 Hz, *J*_2_ = 1.2 Hz, 1H), 7.13 (t, *J* = 8.4 Hz, 2H), 5.74 (s, 2H). ^13^C NMR (150 MHz, DMSO-*d_6_*) δ 153.15, 146.96, 140.77, 135.36, 133.39, 131.54, 130.88, 129.84, 128.82, 125.39, 123.30, 122.49, 120.43, 120.38, 119.40, 118.25, 115.82, 115.68, 115.28, 53.55.

***1-(3-(1-(2-bromobenzyl)-1H-1,2,3-triazol-4-yl)phenyl)-3-(4-methoxyphenyl)urea* (3i)**. White solid. HR MS (ESI) m/z: calcd for C_23_H_21_BrN_5_O_2_ [M+H]^+^478.0879, found 478.0882. ^1^H NMR (600 MHz, DMSO-*d_6_*) δ 8.71 (s, 1H), 8.57 (s, 1H), 8.48 (s, 1H), 8.02 (t, *J* = 1.8 Hz, 1H), 7.71 (d, *J* = 7.8 Hz, 1H), 7.44–7.41 (m, 2H), 7.40–7.36 (m, 3H), 7.33 (t, *J* = 7.8 Hz, 2H), 7.22 (dd, *J*_1_ = 7.2 Hz, *J*_2_ = 1.8 Hz, 1H), 6.89–6.86 (m, 2H), 5.74 (s, 2H), 3.71 (s, 3H). ^13^C NMR (150 MHz, DMSO-*d_6_*) δ 154.98, 153.19, 146.99, 140.93, 139.67, 135.36, 133.39, 133.11, 131.51, 130.89, 129.83, 128.82, 125.39, 123.30, 122.56, 119.56, 118.15, 115.18, 114.45, 55.64, 53.56.

***1-(4-(1-(2-bromobenzyl)-1H-1,2,3-triazol-4-yl)phenyl)-3-(4-methoxyphenyl)urea* (5a)**. White solid. HR MS (ESI) m/z: calcd for C_23_H_21_BrN_5_O_2_ [M+H]^+^478.0879, found 478.0866. ^1^H NMR (600 MHz, DMSO-*d_6_*) δ 8.70 (s, 1H), 8.49 (d, *J* = 11.4 Hz, 2H), 7.76 (d, *J* = 8.4 Hz, 2H), 7.71 (d, *J* = 7.8 Hz, 1H), 7.51 (d, *J* = 9.0 Hz, 2H), 7.43 (t, *J* = 7.2 Hz, 1H), 7.36 (d, *J* = 9.0 Hz, 2H), 7.34–7.31 (m, 1H), 7.21 (d, *J* = 7.2 Hz, 1H), 6.87 (d, *J* = 9.0 Hz, 2H), 5.72 (s, 2H), 3.72 (s, 3H). ^13^C NMR (150 MHz, DMSO-*d_6_*) δ 155.00, 153.11, 147.01, 140.20, 135.36, 133.39, 133.10, 130.89, 128.81, 126.27, 124.46, 123.31, 121.50, 120.56, 118.71, 114.47, 55.65, 53.54.

***1-(4-(1-(3-bromobenzyl)-1H-1,2,3-triazol-4-yl)phenyl)-3-(4-methoxyphenyl)urea* (5b)**. Brown solid. HR MS (ESI) m/z: calcd for C_23_H_21_BrN_5_O_2_[M+H]^+^478.0879, found 478.0887. ^1^H NMR (600 MHz, DMSO-*d_6_*) δ 8.68 (s, 1H), 8.55 (s, 1H), 8.49 (s, 1H), 7.74 (d, *J* = 9.0 Hz, 2H), 7.59 (s, 1H), 7.55 (d, *J* = 7.2 Hz, 1H), 7.51 (d, *J* = 8.4 Hz, 2H), 7.37 (t, *J* = 2.4 Hz, 2H), 7.36–7.35 (m, 2H), 6.87 (d, *J* = 9.0 Hz, 2H), 5.65 (s, 2H), 3.71 (s, 3H). ^13^C NMR (150 MHz, DMSO-*d_6_*) δ 155.00, 153.11, 147.25, 140.20, 139.16, 133.09, 131.53, 131.50, 131.17, 127.51, 126.25, 124.46, 122.35, 121.26, 120.56, 118.72, 114.47, 55.65, 52.61.

***1-(4-(1-(2-bromobenzyl)-1H-1,2,3-triazol-4-yl)phenyl)-3-(4-bromophenyl)urea* (5c)**. White solid. HR MS (ESI) m/z: calcd for C_22_H_18_Br_2_N_5_O [M+H]^+^525.9878, found 525.9870. ^1^H NMR (600 MHz, DMSO-*d_6_*) δ 8.83 (d, *J* = 13.2 Hz, 2H), 8.50 (s, 1H), 7.78 (d, *J* = 8.4 Hz, 2H), 7.71 (dd, *J*_1_ = 8.4 Hz, *J*_2_ = 1.2 Hz, 1H), 7.53 (d, *J* = 8.4 Hz, 2H), 7.46–7.44 (m, 4H), 7.42 (dd, *J*_1_ = 7.2 Hz, *J*_2_ = 1.2 Hz, 1H), 7.33 (td, *J*_1_ = 7.8 Hz, *J*_2_ = 1.2 Hz, 1H), 7.22 (dd, *J*_1_ = 7.8 Hz, *J*_2_ = 1.8 Hz, 1H), 5.72 (s, 2H). ^13^C NMR (150 MHz, DMSO-*d_6_*) δ 152.79, 146.94, 139.77, 139.56, 135.34, 133.39, 131.99, 130.90, 128.81, 126.30, 124.85, 123.32, 121.60, 120.65, 118.96, 113.75, 53.55.

***1-(4-(1-(3-bromobenzyl)-1H-1,2,3-triazol-4-yl)phenyl)-3-(4-bromophenyl)urea* (5d)**. White solid. HR MS (ESI) m/z: calcd for C_22_H_18_Br_2_N_5_O [M+H]^+^525.9878, found 525.9881. ^1^H NMR (600 MHz, DMSO-*d_6_*) δ 8.86 (s, 1H), 8.83 (s, 1H), 8.56 (s, 1H), 7.76 (d, *J* = 9.0 Hz, 2H), 7.59 (s, 1H), 7.56 (dt, *J*_1_ = 7.2 Hz, *J*_2_ = 1.8 Hz, 1H), 7.53 (d, *J* = 9.0 Hz, 2H), 7.45 (s, 4H), 7.38–7.34 (m, 2H), 5.65 (s, 2H). ^13^C NMR (150 MHz, DMSO-*d_6_*) δ 152.79, 147.18, 139.79, 139.56, 139.14, 131.99, 131.53, 131.50, 131.18, 127.52, 126.27, 124.84, 122.35, 121.36, 120.65, 118.97, 113.74, 52.62.

***1-(4-bromophenyl)-3-(4-(1-(3-methoxybenzyl)-1H-1,2,3-triazol-4-yl)phenyl)urea* (5e)**. White solid. HR MS (ESI) m/z: calcd for C_23_H_21_BrN_5_O_2_ [M+H]^+^478.0879, found 478.0886. ^1^H NMR (600 MHz, DMSO-*d_6_*) δ 8.88 (s, 1H), 8.84 (s, 1H), 8.53 (s, 1H), 7.76 (d, *J* = 9.0 Hz, 2H), 7.52 (d, *J* = 8.4 Hz, 2H), 7.45 (s, 4H), 7.31 (t, *J* = 7.8 Hz, 1H), 6.95 (s, 1H), 6.93–6.89 (m, 2H), 5.59 (s, 2H), 3.75 (s, 3H). ^13^C NMR (150 MHz, DMSO-*d_6_*) δ 159.95, 152.80, 147.10, 139.74, 139.58, 137.96, 131.99, 130.44, 126.24, 124.94, 121.21, 120.64, 120.45, 118.97, 114.20, 113.96, 113.73, 55.60, 53.39.

***1-(4-(1-(2-bromobenzyl)-1H-1,2,3-triazol-4-yl)phenyl)-3-(4-chlorophenyl)urea* (5f)**. White solid. HR MS (ESI) m/z: calcd for C_22_H_18_BrClN_5_O [M+H]^+^482.0383, found 482.0374. ^1^H NMR (600 MHz, DMSO-*d_6_*) δ 8.86 (d, *J* = 13.2 Hz, 2H), 8.52 (s, 1H), 7.78 (d, *J* = 9.0 Hz, 2H), 7.72 (dd, *J*_1_ = 7.8 Hz, *J*_2_ = 1.2 Hz, 1H), 7.53 (d, *J* = 8.4 Hz, 2H), 7.52-7.49(m, 2H), 7.44 (td, *J*_1_ = 7.8 Hz, *J*_2_ = 1.2 Hz, 1H), 7.35-7.32(m, 3H), 7.22 (dd, *J*_1_ = 7.8 Hz, *J*_2_ = 1.8 Hz, 1H), 5.73 (s, 2H). ^13^C NMR (150 MHz, DMSO-*d_6_*) δ 152.82, 146.93, 139.79, 139.68, 139.12, 135.35, 133.39, 130.89, 129.12, 128.82, 126.29, 125.85, 125.40, 124.81, 123.31, 121.61, 120.22, 118.93, 53.54, 34.87, 30.88, 21.52.

***1-((1-(2-bromobenzyl)-1H-1,2,3-triazol-4-yl)methyl)-3-(4-methoxyphenyl)urea* (7a)**. Gray solid. HR MS (ESI) m/z: calcd for C_18_H_19_BrN_5_O_2_[M+H]^+^416.0722, found 416.0714. ^1^H NMR (600 MHz, DMSO-*d_6_*) δ 8.34 (s, 1H), 7.97 (s, 1H), 7.69 (d, *J* = 7.8 Hz, 1H), 7.40 (td, *J*_1_ = 7.2 Hz, *J*_2_ = 1.2 Hz, 1H), 7.31 (td, *J*_1_ = 7.2 Hz, *J*_2_ = 1.2 Hz, 1H), 7.28 (d, *J* = 8.4 Hz, 2H), 7.14 (dd, *J*_1_ = 7.2 Hz, *J*_2_ = 1.2 Hz, 1H), 6.81 (d, *J* = 9.0 Hz, 2H), 6.47 (t, *J* = 5.4 Hz, 1H), 5.66 (s, 2H), 4.32 (d, *J* = 5.4Hz, 2H), 3.69 (s, 3H). ^13^C NMR (150 MHz, DMSO-*d_6_*) δ 155.78, 154.47, 135.51, 133.94, 133.37, 130.85, 130.83, 128.74, 123.67, 123.30, 119.94, 114.35, 55.61, 53.29, 35.36.

***1-((1-(3-bromobenzyl)-1H-1,2,3-triazol-4-yl)methyl)-3-(4-methoxyphenyl)urea* (7b)**. Gray solid. HR MS (ESI) m/z: calcd for C_18_H_19_BrN_5_O_2_[M+H]^+^416.0722, found 416.0729. ^1^H NMR (600 MHz, DMSO-*d_6_*) δ 8.34 (s, 1H), 8.08 (s, 1H), 7.53 (d, *J* = 9.6 Hz, 2H), 7.34 (t, *J* = 7.8 Hz, 1H), 7.31 (d, *J* = 7.8 Hz, 1H), 7.28 (d, *J* = 9.0 Hz, 2H), 6.81 (d, *J* = 9.0 Hz, 2H), 6.47 (s, 1H), 5.59 (s, 2H), 4.31 (s, 2H), 3.69 (s, 3H). ^13^C NMR (150 MHz, DMSO-*d_6_*) δ 155.85, 154.47, 139.25, 133.94, 131.48, 131.43, 131.20, 127.56, 123.74, 122.28, 119.94, 114.34, 55.61, 52.37, 35.43.

***1-((1-(3-methoxybenzyl)-1H-1,2,3-triazol-4-yl)methyl)-3-(4-methoxyphenyl)urea* (7c)**. White solid. HR MS (ESI) m/z: calcd for C_19_H_22_N_5_O_3_ [M+H]^+^368.1723, found 368.1731. ^1^H NMR (600 MHz, DMSO-*d_6_*) δ 8.34 (s, 1H), 8.03 (s, 1H), 7.29-7.26 (m, 3H), 6.89 (d, *J* = 6.0 Hz, 2H), 6.86 (d, *J* = 7.2 Hz, 1H), 6.81 (d, *J* = 9.0 Hz, 2H), 6.46 (s, 1H), 5.53 (s, 2H), 4.31 (s, 2H), 3.73 (s, 3H), 3.69 (s, 3H). ^13^C NMR (150 MHz, DMSO-*d_6_*) δ 159.88, 155.84, 154.47, 138.02, 133.93, 130.36, 120.50, 119.93, 114.34, 114.24, 113.87, 55.60, 55.56, 53.16, 35.42.

***1-((1-(2-bromobenzyl)-1H-1,2,3-triazol-4-yl)methyl)-3-(4-bromophenyl)urea* (7d)**. White solid. HR MS (ESI) m/z: calcd for C_17_H_16_Br_2_N_5_O [M+H]^+^463.9722, found 463.9729. ^1^H NMR (600 MHz, DMSO-*d_6_*) δ 8.71 (s, 1H), 7.99 (s, 1H), 7.68 (d, *J* = 7.8 Hz, 1H), 7.41–7.35 (m, 5H), 7.31 (td, *J*_1_ = 7.8 Hz, *J*_2_ = 1.2 Hz, 1H), 7.15 (d, *J* = 6.6 Hz, 1H), 6.64 (t, *J* = 4.8 Hz, 1H), 5.66 (s, 2H), 4.34 (d, *J* = 5.4 Hz, 2H). ^13^C NMR (150 MHz, DMSO-*d_6_*) δ 155.36, 140.27, 135.49, 133.37, 131.83, 130.86, 130.83, 128.74, 123.76, 123.30, 120.08, 112.90, 53.31, 35.35.

***1-((1-(3-bromobenzyl)-1H-1,2,3-triazol-4-yl)methyl)-3-(4-bromophenyl)urea* (7e)**. White solid. HR MS (ESI) m/z: calcd for C_17_H_16_Br_2_N_5_O [M+H]^+^463.9722, found 463.9727. ^1^H NMR (600 MHz, DMSO-*d_6_*) δ 8.71 (s, 1H), 8.06 (s, 1H), 7.53 (d, *J* = 8.4 Hz, 2H), 7.39–7.35 (m, 4H), 7.33 (d, *J* = 8.4 Hz, 1H), 7.31 (d, *J* = 7.2 Hz, 1H), 6.64 (s, 1H), 5.59 (s, 2H), 4.33 (d, *J* = 4.8 Hz, 2H). ^13^C NMR (150MHz, DMSO-*d_6_*) δ 155.36, 140.27, 139.26, 131.82, 131.48, 131.43, 131.19, 127.56, 123.50, 122.28, 120.08, 112.90, 52.35, 35.36.

***1-(4-bromophenyl)-3-((1-(3-methoxybenzyl)-1H-1,2,3-triazol-4-yl)methyl)urea* (7f)**. Brown solid. HR MS (ESI) m/z: calcd for C_18_H_19_BrN_5_O_2_[M+H]^+^416.0722, found 416.0715. ^1^H NMR (600 MHz, DMSO-*d_6_*) δ 8.70 (s, 1H), 8.00 (s, 1H), 7.39–7.36 (m, 4H), 7.30–7.25 (m, 1H), 6.90–6.88 (m, 2H), 6.86 (d, *J* = 7.8 Hz, 1H), 6.63 (t, *J* = 5.4 Hz, 1H), 5.53 (s, 2H), 4.32 (d, *J* = 5.4Hz, 2H), 3.73 (s, 3H). ^13^C NMR (150 MHz, DMSO-*d_6_*) δ 159.89, 155.34, 140.27, 138.06, 131.83, 130.36, 123.27, 120.50, 120.07, 114.24, 113.86, 112.89, 55.56, 53.11, 35.35.

^1^H NMR spectrum of compounds **3a**–**3i**, **5a**–**5f** and **7a**–**7f** are shown in Appendix A.

### 3.4. IDO1 Enzymatic Inhibition Assay

To demonstrate the inhibitory effects of the designed compounds against IDO1, the Hela cell density was adjusted using DMEM complete medium and subsequently seeded into 96-well cell culture plates at 100 μL per well for a total of 50,000 cells, and then incubated at 37 °C, 5% CO_2_ overnight. The next day, 100 μL of various concentrations of the compounds were diluted in a medium containing 100 ng/mL human IFNγ, added onto the 96-well plate, then incubated for 18 h. On the third day, 140 μL of the medium was removed from a new 96-well plate and 10 μL of 6.1 N TCA was added to precipitate the protein at 50 °C; for 30 min. The sediment was centrifuged at 2500 rpm for 10 min. Then, the supernate was transferred to another 96-well plate and mixed with 100 μL (2% (*w*/*v*)) of 4-(dimethylamino) benzaldehyde in acetic acid. The plate was incubated at room temperature for 10 min, and the yellow color derived from the kynurenine was recorded by measuring absorbance at 480 nm using a microplate reader (PE, Envision; Perkinelmer & Co.: Wellesley, MA, USA, 2019). Graphs of inhibition curves with IC_50_ values were generated using Prism 6.0 [2,22].

### 3.5. Molecular Docking 

Docking experiments were performed by using Schrödinger (LLC 2015; Software for Technical Computation; Schrödinger Inc.: New York, NY, USA, 2015), and 4PK5 with Amg-1 as the co-crystal ligand was chosen as the complex system. Protein was processed and optimized by the Protein Preparation Wizard, which included hydrogenation of protein, assigned bond orders, creation of zero-order bonds with metals, creation of disulfide bonds, removal of water molecules larger than 5 Å from het groups, structural optimization, and energy minimization. The ligands were constructed by ChemDraw (version 16.0; Software for Technical Computation; Perkinelmer & Co.: Wellesley, MA, USA, 2016) and optimized by the LigPrep module under the condition of pH 7.0 ± 2.0 for protonation and generation of stereoisomers. Molecular lattices were generated centered on Amg-1, while metal coordination constraints centered on heme iron were generated; docking was performed using SP precision. The 3D binding patterns of compounds to IDO1 were visualized by PyMOL (version 2.4.0; Software for Technical Computation; Delano Scientific LLC: Silicon Valley, CA, USA, 2012).

### 3.6. Quantum Mechanical Studies (QM)

To better understand the molecular properties, we performed density functional theory (DFT) studies on compounds **3a**, **5e** and **7f** based on quantum chemistry [23]. DFT is used to study the electronic structure of multi-electron systems, which has been widely used to calculate the electronic properties of molecules. The DFT calculation was implemented by Gaussian09 software (Software for Technical Computation; Gaussian Inc.: PIT, USA, 2017). Geometry optimization of compounds wasperformed by using the B3LYP/6–31G (d,p) basis sets [24]. The B3LYP/6–311G (d,p) basis sets were applied for the frontier orbital and molecular electrostatic potential (MEP)analysis. The highest occupied molecular orbital (HOMO) energy, the lowest unoccupied molecular orbital (LUMO) energy, and the energy gap ($\Delta E_{\mathrm{LUMO}-\mathrm{HOMO}}$) parameters were investigated.

## 4. Conclusions

In this study, three series of compounds containing a urea structure and 1,2,3-triazole structure were designed, among which, the compound **3** series exhibited better inhibitory effects on IDO1, especially compound **3a** (IC_50_ = 0.75 μM). Molecular docking and quantum mechanical studies showed that compound **3a** was more chemically reactive than **5e** and **7f** with the same substituents, further demonstrating the potential activity of compound **3a**. Therefore, we will optimize the compound **3** series and carry out experimental studies at the cellular and animal levels in the future.

## Figures and Tables

**Figure 1 pharmaceuticals-15-01316-f001:**
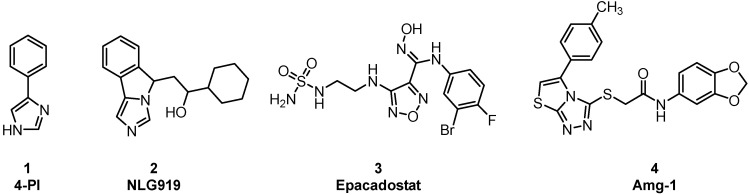
Chemical structures of four IDO1 inhibitors.

**Figure 2 pharmaceuticals-15-01316-f002:**
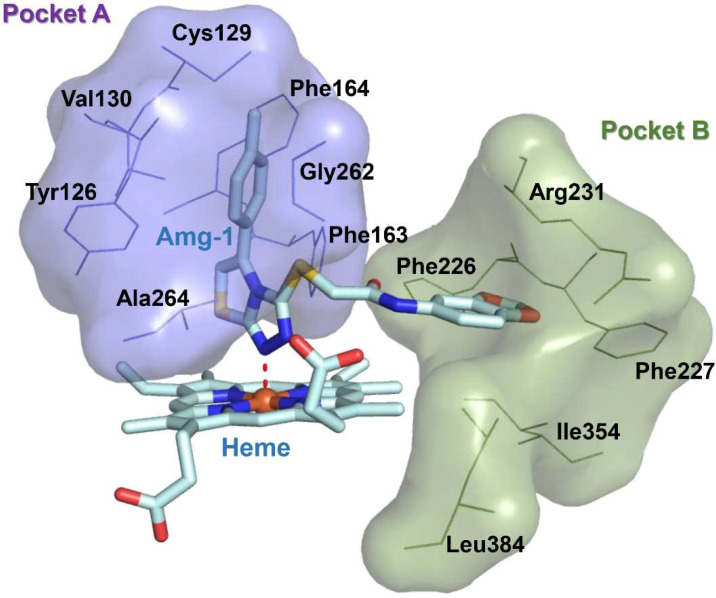
Pocket surface map of the crystal structure of IDO1 complexed with Amg-1 (PDB code: 4PK5). Purple represents amino acids contained in pocket A; green represents amino acids contained in pocket B.

**Figure 3 pharmaceuticals-15-01316-f003:**
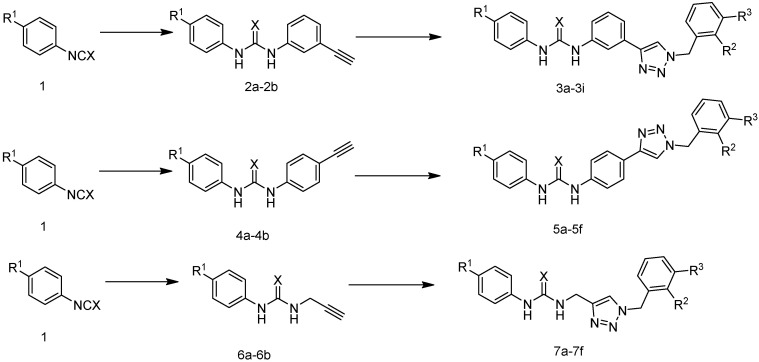
The reaction routes to compounds **3a**–**3i**, **5a**–**5f**, and **7a**–**7f**.

**Figure 4 pharmaceuticals-15-01316-f004:**
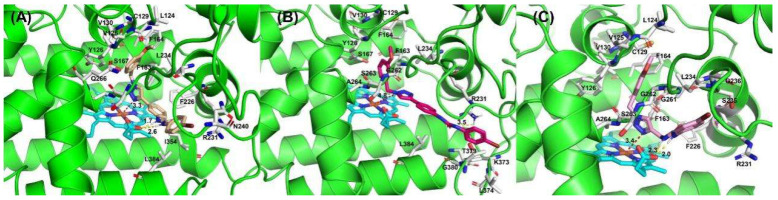
The binding modes of compounds **3a** (**A**), **5e** (**B**) and **7f** (**C**).

**Figure 5 pharmaceuticals-15-01316-f005:**
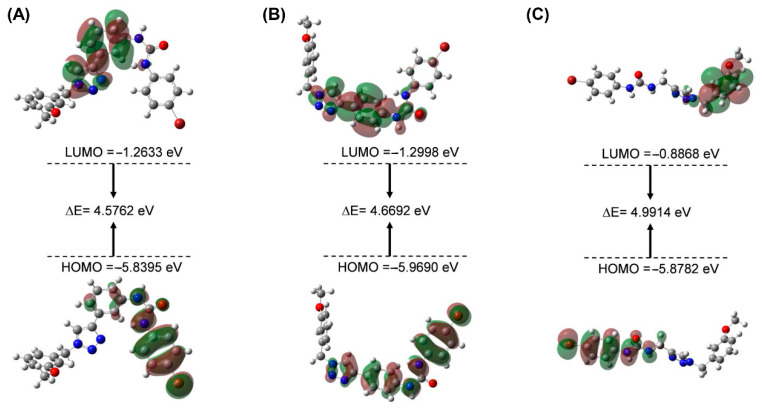
Frontier molecular orbitals of compounds **3a** (**A**), **5e** (**B**) and **7f** (**C**).

**Figure 6 pharmaceuticals-15-01316-f006:**
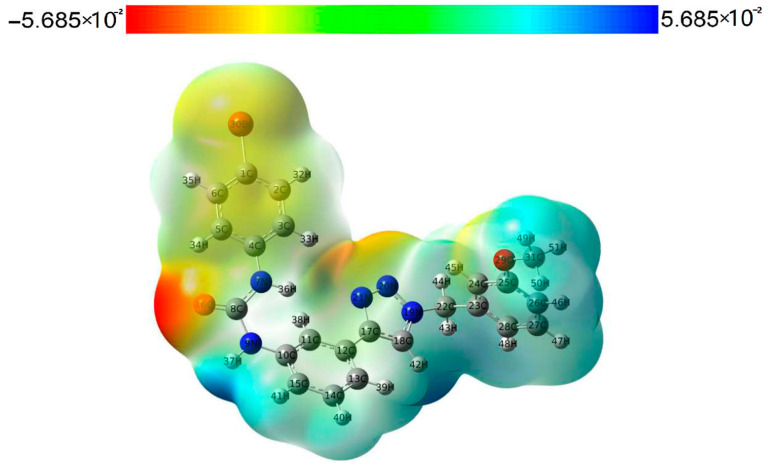
MEP surfaces of compound **3a**.

**Table 1 pharmaceuticals-15-01316-t001:** R-group of compounds **3a**–**3i**, **5a**–**5f**, and **7a**–**7f**.

Compound	X	R^1^	R^2^	R^3^	Yield (%)
**3a**	O	Br	H	OCH_3_	73.9
**3b**	O	OCH_3_	H	OCH_3_	55.1
**3c**	O	F	H	OCH_3_	80.8
**3d**	S	H	H	OCH_3_	69.2
**3e**	O	Br	H	Br	85.9
**3f**	O	OCH_3_	H	Br	77.4
**3g**	O	Br	Br	H	86.6
**3h**	O	F	Br	H	63.8
**3i**	O	OCH_3_	Br	H	49.1
**5a**	O	OCH_3_	Br	H	41.7
**5b**	O	OCH_3_	H	Br	87.2
**5c**	O	Br	Br	H	39.4
**5d**	O	Br	H	Br	33.4
**5e**	O	Br	H	OCH_3_	40.2
**5f**	O	Cl	Br	H	76.4
**7a**	O	OCH_3_	Br	H	36.6
**7b**	O	OCH_3_	H	Br	43.9
**7c**	O	OCH_3_	H	OCH_3_	39.8
**7d**	O	Br	Br	H	43.2
**7e**	O	Br	H	Br	86.9
**7f**	O	Br	H	OCH_3_	45.4

**Table 2 pharmaceuticals-15-01316-t002:** IDO1 inhibitory activities of designed derivatives.

Compd No.	IC_50_ (μM)	Compd No.	IC_50_ (μM)	Compd No.	IC_50_ (μM)
	IDO1		IDO1		IDO1
**3a**	0.75 ± 0.27	**5a**	>10	**7a**	>10
**3b**	3.46 ± 0.71	**5b**	7.43 ± 1.05	**7b**	3.97 ± 0.92
**3c**	6.21 ± 0.89	**5c**	>10	**7c**	7.10 ± 1.48
**3d**	4.13 ± 0.42	**5d**	>10	**7d**	>10
**3e**	8.16 ± 1.12	**5e**	>10	**7e**	>10
**3f**	5.93 ± 0.97	**5f**	>10	**7f**	>10
**3g**	2.59 ± 0.13				
**3h**	2.58 ± 0.59				
**3i**	0.80 ± 0.37				

IC_50_ values were fitted from single-point inhibition curves, and two parallel experiments were performed for each compound. IC_50_ values were calculated using Graph Pad Prism 6.0 software. These results are reported as the averages ± SD.

**Table 3 pharmaceuticals-15-01316-t003:** Molecular frontier orbital parameters of compounds **3a**, **5e** and **7f**.

Compd No.	HOMO Energy (eV)	LUMO Energy (eV)	$\Delta E_{\mathrm{LUMO}-\mathrm{HOMO}}\;(\mathrm{eV})$
**3a**	−5.8395	−1.2633	4.5762
**5e**	−5.9690	−1.2998	4.6692
**7f**	−5.8782	−0.8868	4.9914

## Data Availability

Data are contained within the article.

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
