# Peer review of "Discovery of Novel 1,2,3-triazole Derivatives as IDO1 Inhibitors"

_pharmaceuticals, 2022, doi:10.3390/ph15111316_

Round 1
Reviewer 1 Report
The manuscript by Yang and coworkers, describes the synthesis of 1,2,3-triazole derivatives connected to urea moieties with different linkers and their evaluation as IDO1 inhibitors. Indoleamine 2,3-dioxygenase 1 is an interesting target in the development of potential chemotherapeutic agents. Compound 3a shows a good IC50 value of 0.75 μM. In addition, the authors describe molecular docking and quantum mechanical studies.
The work is clearly presented, the experimental procedures are carefully detailed, and the conclusions are reasonably supported by the data.
Regarding the experimental part, although there is no doubt about the identity of the reported substrates the full characterization of the new compounds should include not only 1H and 13C NMR spectra, but also either HRMS or elemental analysis. Moreover, reporting the physical state of the compounds (white/yellow/orange solid/oil) is very convenient for all compounds and the melting point for solids is also advisable.
Although I am not a native English, I could find many grammatical and orthographical mistakes and typos in the whole manuscript. Some examples:
The numbers of the compounds should be formatted in bold throughout the text in order to avoid confusion with other numbers.
The numbering of the substituents R1, R2, R3… should be formatted in superscripts, since a subscript may mean several R (I.e R2 means two R).
m (meta), p (para) should be formatted in bold
Page 3, paragraph 1 “compounds are” not “compoundsare”; paragraph 2 “all target” not “alltarget” (space missing)
Page 4, “OCH3” not “OCH3” (subscript)
Page 7, “The 3.05 Ntrichloroacetic acid” (spaces missed)
Spaces should be inserted before the square brackets in references and between “=” symbol
I advise a careful check throughout the whole manuscript and a revision of the redaction.
In summary, I think that the authors have demonstrated the value of their research and they provide a nice contribution to the field of click chemistry and the development of chemotherapeutic agents that may be of interest for Medicinal an Organic Chemists. Therefore, I support the publication of the manuscript in Pharmaceuticals after the issues commented in the report are taken into account.
Reviewer 2 Report
In this submitted paper, evidently the synthetic part is the least relevant. However, it should be dealt with more thoroughly. Figure 3 shows the scheme of the synthetic process and it is not clear whether adducts 2,4 and 6 are isolated reaction intermediates or if the overall reaction takes place in a single step. The experimental conditions must be better detailed: it would be appropriate to indicate them in figure 3. The experimental part is extremely lacking and written in a hasty and inadequate manner. Finally, the yields with which the various adducts were obtained are never reported and it would be appropriate to indicate them in Table 1
The authors claim to have synthesized new products: if so, characterization with only NMR is not sufficient. If solid, the melting point must be reported. The elemental analysis or, alternatively, the exact mass must also be reported.
I am not able to refer to the pharmacological section; however I noticed that the references are not reported in an appropriate manner. The authors indicate 21 references but then only 17 are found in the text. Moreover in page 6 other references are probably cited without, however, any matching in the list of references at the end of the manuscript.
Round 2
Reviewer 2 Report
The manuscript now can be published
Author Response
Dear reviewer,
Thank you so much for your careful check. According to your suggestion, we have revised the format and put the numbers included in the chemical formulas into subscripts for the HR MS description.